# Secondary Metabolites from the Genus *Eurotium* and Their Biological Activities

**DOI:** 10.3390/foods12244452

**Published:** 2023-12-12

**Authors:** Jiantianye Deng, Yilong Li, Yong Yuan, Feiyan Yin, Jin Chao, Jianan Huang, Zhonghua Liu, Kunbo Wang, Mingzhi Zhu

**Affiliations:** 1National Research Center of Engineering and Technology for Utilization of Botanical Functional Ingredients, Hunan Agricultural University, Changsha 410128, China; tianye071216@163.com (J.D.); yllicn@hotmail.com (Y.L.); jian7513@sina.com (J.H.); zhonghua-liu@hotmail.com (Z.L.); wkboo163@163.com (K.W.); 2Co-Innovation Center of Education Ministry for Utilization of Botanical Functional Ingredients, Hunan Agricultural University, Changsha 410128, China; 3Key Laboratory of Tea Science of Ministry of Education, Hunan Agricultural University, Changsha 410128, China; 4Hunan Tea Group Co., Ltd., Changsha 410128, China; qwehyx1999@gmail.com (Y.Y.); l1553193716@gmail.com (F.Y.); bohao-shang0122@hotmail.com (J.C.)

**Keywords:** *Eurotium*, *Eurotium cristatum*, secondary metabolites, anthraquinones, benzaldehyde derivatives, biological activity

## Abstract

*Eurotium* is the teleomorph genus associated with the section *Aspergillus. Eurotium* comprises approximately 20 species, which are widely distributed in nature and human environments. *Eurotium* is usually the key microorganism for the fermentation of traditional food, such as Fuzhuan brick tea, Liupao tea, Meju, and Karebushi; thus, *Eurotium* is an important fungus in the food industry. *Eurotium* has been extensively studied because it contains a series of interesting, structurally diverse, and biologically important secondary metabolites, including anthraquinones, benzaldehyde derivatives, and indol diketopiperazine alkaloids. These secondary metabolites have shown multiple biological activities, including antioxidative, antimicrobial, cytotoxic, antitumor, insecticidal, antimalarial, and anti-inflammatory activities. This study presents an up-to-date review of the phytochemistry and biological activities of all *Eurotium* species. This review will provide recent advances on the secondary metabolites and their bioactivities in the genus *Eurotium* for the first time and serve as a database for future research and drug development from the genus *Eurotium*.

## 1. Introduction

*Eurotium* (Eurotiaceae), now renamed *Aspergillus*, is the sexual generation of the genus *Aspergillus*. Despite the renaming, the majority of mycologists prefer adhering to the established and commonly used nomenclature [1,2,3]. *Eurotium* is characterised by its golden cleistothecia, lenticular ascospores, uniseriate conidial heads in shades of green or blue, and yellow-, orange- or red-encrusted hyphae [2,4]. The genus *Eurotium* comprises approximately 20 species [2], of which *Eurotium amstelodami*, *Eurotium cristatum*, and *Eurotium repens* have received the most attention [5,6]. All species of *Eurotium* are hypertonic fungi, which are widely distributed in nature and human environments, especially in environments of high salt, high sugar, and low water. *Eurotium* species are generally considered to be benign fungi without mycotoxins [7,8,9,10,11]. Therefore, *Eurotium* species are widely used in the food processing industry. Katsuobushi is a traditional Japanese food made from tuna fermented by *Eurotium*. During the fermentation process, the *Eurotium* reduces the fat content of the tuna and turns it to a deep red colour, giving it a milder taste and a unique flavour [12]. Meju is a traditional Korean fermented soybean product, and the dominant fungus at the middle and late stages of its fermentation is *Eurotium*. During the fermentation process, the microorganisms break down the large nutrients in soybeans to form small molecules such as amino acids, small peptides, urea cycle intermediates, nucleosides, and organic acids, resulting in its unique organoleptic qualities and health activities [13]. In addition, in the fermentation process of China’s traditional beverage Fuzhuan brick tea, *Eurotium* is the most dominant strain, accounting for more than 98% of all fermentation microorganisms. *Eurotium* breaks down a variety of compounds in the tea and forms products such as free amino acids, polyphenolic compounds, purine alkaloids, and terpenoids, reducing the bitter and astringent flavour, as well as giving the tea a unique ‘fungal flower’ aroma [14]. Functioning as key microorganisms in these processes, *Eurotium* contributes to the degradation of complex molecules into smaller, nutritionally rich compounds by secreting microbial enzymes. These secondary metabolites not only impart distinctive flavours to fermented foods but also enhance their potential health benefits.

The exploration of secondary metabolites within *Eurotium* has a rich history, dating back to the 19th century when the chemical structure of *Eurotium*’s pigments was first identified. Substantial advancements in understanding *Eurotium*’s secondary metabolites have been achieved in recent decades [5,6]. Notably, marine environments and fermented food and drink have become important sources of *Eurotium* species in recent years, leading to the identification of numerous novel secondary metabolites [15]. The compounds isolated from *Eurotium* species mainly include anthraquinones, benzaldehyde derivatives, and indol diketopiperazine alkaloids. These secondary metabolites exhibit various bioactivities, such as antioxidative, antimicrobial, cytotoxic, antitumor, insecticidal, antimalarial, and anti-inflammatory activities [14,16,17,18,19,20,21]. These physiologically active secondary metabolites are simultaneously ingested by people along with fermented foods or beverages, resulting in effects on human health. However, existing studies have mainly focused on food products related to the fermentation of *Eurotium*, and there is no review article in English that systematically summarises the secondary metabolites and physiological activities of *Eurotium*.

In this context, this review will provide the recent advances on the secondary metabolites and their bioactivities in the genus *Eurotium* for the first time (Table 1). Meanwhile, the review outlines future prospects and challenges with a view to providing new insights into the development of relevant fermented foods.

## 2. Secondary Metabolites from *Eurotium*

Nearly 180 compounds have been isolated and identified from *Eurotium* species using nuclear magnetic resonance (NMR) spectroscopy. These compounds mainly include anthraquinones, benzaldehyde derivatives, and indol diketopiperazine alkaloids. These secondary metabolites are not only derived from food but also produced by some *Eurotium* species in other environments, and we have included them in this review for future use in the fermented food industry.

### 2.1. Anthraquinones

Anthraquinones, which are formed by the merger of three benzene rings, are the largest group of natural pigments of quinoids [22]. Typically synthesised by plants and microorganisms, anthraquinones contribute hues (usually yellow, orange, or brown) to lichens, as well as the mycelium and fruiting bodies of fungi [23]. Fungal anthraquinones commonly feature several side substituents on the benzene ring, with 1,8-dihydroxy and 1,5,8 or 1,6,8 trihydroxy anthraquinone derivatives being prevalent [69]. Anthraquinones have shown a variety of pharmacological activities, including antibacterial, antiviral, insecticidal, diuretic, diarrhoeal, immunomodulatory, and anticancer effects [11,27,70].

The exploration of *Eurotium* anthraquinones commenced in 1980, spearheaded by Anke et al. [22]. Their comprehensive investigation encompassed the structural analysis of pigments in 20 *Eurotium* species, including *Eurotium aeutum*, *Eurotium glabrum*, *Eurotium herbariorum*, *Eurotium pseudoglaucum*, *E. repens*, *Eurotium rubrum*, *Eurotium tonophihtm*, *Eurotium umbrosum*, *Eurotium appendiculatum*, *Eurotium carnoyi*, *Eurotium echinulatum*, *Eurotium niveoglaucum*, *E. amstelodami*, *Eurotium chevalieri*, *E. cristatum*, *Eurotium heterocaryoticum*, *Eurotium intermedium*, *Eurotium leucocarpum*, *Eurotium montevidensis*, and *Eurotium spiculosum*. They found that these pigments were polyhydroxy anthraquinones, including questin (**1**), physcion (**2**), erythroglaucin (**3**), emodin (**4**), catenarin (**5**), rubrocristin (**6**), rubrocristin-8-methylether (**7**), rubrocristin-6-acetate (**8**), and querstin-6-methylether (**9**). Further, rubrocristin, a new yellow pigment, was first discovered in nature. The production of these pigments was seriously affected by the concentrations of glucose and salt in culture medium. It has been proved that the number of hydroxyl groups and their position play an essential role in the antibacterial activity of these polyhydroxy anthraquinones. In addition, physcion was supposed to play a role in iron transport or the metabolism of fungal cells [71]. Three anthraquinones, including 2-*O*-methyleurotinone (**10**), 2,12-dimethyleurotinone (**11**), and eurotinone (**12**), were isolated from *E. echinulatum* by Eder et al. [29] These compounds exhibited antiangiogenic effects, suggesting their potential applications in preventing and treating malignant diseases. Another strain of interest, *E. herbariorum* NU-2, isolated during the manufacturing of Karebushi (a traditional food in Japan), was investigated by Miyake et al. [16], who then identified physcion-10,10′-bianthrone (**13**), questinol (**14**), asperflavin (**15**), as well as questin, physcion, and catenarin, in this fungus. Additionally, some pigments of anthraquinones, including variecolorquinone A (**16**), questin, physcion, erythroglaucin, emodin, catenarin, questinol, and asperflavin, were also found in other *Eurotium* strains, such as *Eurotium* sp. M30 XS-2012 [11] or *E. cristatum* KUFC 7356 [26].

The exploration of bioactive compounds in marine microorganisms has garnered considerable attention in recent years [1]. Li et al. [23] isolated the *E. rubrum* strain from a marine mangrove plant *Hibiscus Tiliaceus*, and then identified four new anthraquinones, as well as three known anthraquinones (questin, 2-*O*-methyleurotinone, and asperflavin) in this fungus. These four new anthraquinones were 2-*O*-methyl-4-*O*-(*α*-_D_-ribofuranosyl)-9-dehydroxyeurotinone (**17**; colourless amorphous powder), 2-*O*-methyl-9-dehydroxyeurotinone (**18**; colourless amorphous powder), eurorubrin (**19**; brown amorphous powder), and 3-*O*-(*α*-_D_-ribofuranosyl)-questin (**20**; orange amorphous powder). Based on the spectral data, 2-*O*-methyl-9-dehydroxyeurotinone is a 9-dehydroxyl derivative of 2-*O*-methyleurotinone; eurorubrin is a symmetrical dimeric compound composed of two molecules of asperflavin via a methylene group; 3-*O*-(*α*-_D_-ribofuranosyl)-questin is a glycoside consisting of questin as aglycone and one sugar unit. In addition, Du et al. [31] isolated an endophytic fungus (*E. cristatum* EN-220) from the marine alga *Sargassum thunbergii*, and identified one new anthraquinone glycoside named 3-*O*-(*α*-_D_-ribofuranosyl)-questinol (**21**; red amorphous powder), as well as asperflavin ribofuranoside (**22**), asperflavin, (+)-variecolorquinone A, eurorubrin, and 3-*O*-(*α*-_D_-ribofuranosyl)-questin. 3-*O*-(*α*-_D_-ribofuranosyl)-questinol and 3-*O*-(*α*-_D_-ribofuranosyl)-questin have the same ribose residue. A new anthraquinone named 9-dehydroxyeurotinone (**23**; colourless amorphous powder) was also found in *E. rubrum* [28]. Zin et al. [20] isolated a new compound named acetylquestinol (**24**; yellow crystal), as well as four known anthraquinones, questin, physcion, emodin, and questinol, from the culture of the mangrove plant *Rhizophora mucronata*-derived endophytic fungus *E. chevalieri* KUFA 0006. Acetylquestinol is a 1,3,6,8-tetrasubsituted 9,10-anthraquinone, similar to questinol. Further, the metabolites vary greatly between the *E. chevalieri* KUFA 0006 and soil-derived strain of *E.chevalier* [58]. Additionally, questinol was also isolated from the marine-derived *E. amstelodami* [30].

The endophytes derived from saline-alkali plants are attracting increasing attention due to the extreme environment of high osmolarity and nutrient deprivation. The chemical investigation of saline-alkali plant-derived endophytic fungi has just begun compared with those of marine mangrove plant-derived endophytes. Zhang et al. [27] found a new anthraquinone named rubrumol (**25**), as well as emodin, catenarin, rubrocristin, and 2-*O*-methyleurotinone, in a halo-tolerant endophytic fungus *E. rubrum.* This fungus is derived from the salt-tolerant wild plant *Suaeda salsa.* These anthraquinones displayed topoisomerase inhibitory activity, which implied that endophytic *Eurotium* fungi from saline-alkali plants may be one new reservoir for natural products in the future (Figure 1).

### 2.2. Benzaldehyde Derivatives

Benzaldehyde derivatives constitute a class of polyketides synthesised via the combination of polyketone and terpenoid pathways [40]. It has been reported that benzaldehyde derivatives have various bioactivities, including antioxidative, antibacterial, antifungal, antitumor, antimalarial, and antileishmanial activities [33,36,38]. Benzaldehyde derivatives, which are a kind of natural pigments, are a class of main metabolites in the genus *Eurotium* [14]. Over 20 benzaldehyde derivatives have been identified in *Eurotium*. 

Four new and seven known benzaldehyde derivatives were identified from *E. rubrum*, an endophytic fungus isolated from the inner tissue of stems in the mangrove plant *Hibiscus tiliaceus* by Li et al. [33] These four benzaldehyde derivatives were 2-(2′,3-epoxy-1′-heptenyl)-6-hydroxy-5-(3″-methyl-2″-butenyl)-benzaldehyde (**26**; yellowish amorphous powder), (*E*)-6-hydroxy-7-(3-methyl-2-butenyl)-2-(3-oxobut-1-enyl)-chroman-5-carbaldehyd (**27**; yellowish amorphous powder), 2-(1′,5′-heptadienyl)-3,6-dihydroxy-5-(3″-methyl-2″-butenyl)-benzaldehyde (**28**; yellowish amorphous powder), and eurotirumin (**29**; yellowish amorphous powder). The seven known benzaldehyde derivatives were chaetopyranin (**30**), flavoglaucin (**31**), aspergin (**32**), isotetrahydroauroglaucin (**33**), isodihydroauroglaucin (**34**), 2-(2′,3-epoxy-1′,3′-heptadienyl)-6-hydroxy-5-(3-methyl-2-butenyl)-benzaldehyde (**35**), and 2-(2′,3-epoxy-1′,3′,5′-heptatrienyl)-6-hydroxy-5-(3-methyl-2-butenyl)-benzaldehyde (**36**). These four benzaldehyde derivatives possess a penta-substituted benzene ring system bearing a 3-methyl-2-butenyl at C-5 and a phenolic hydroxyl group at C-6. The structures of compounds **26** and **35** are similar, except that two olefinic carbon signals of C-3′ and C-4′ in the ^13^C-NMR of compound **35** are replaced by two methylene signals at C-3′ and C-4′ in compound **26**. The structures of compounds **27** and **30** are similar, except that the signals at H-6′ and C-6′ in compound **30** are replaced by a carbonyl signal at C-6′ in compound **27**. The structures of compounds **28** and **34** are similar, and the inconsistent position for the two double bonds in the heptadienyl side chain is the only difference. Li et al. [39] also isolated two new benzaldehyde derivatives, eurotirubrin A (**37**) and eurotirubrin B (**38**; yellow powder) in *E. rubrum* in another research work. In addition, auroglaucin (**39**), tetrahydroauroglaucin (**40**), dihydroauroglaucin (**41**), flavoglaucin, and isodihydroauroglaucin were identified from Karebushi-derived *Eurotium* fungi. All four benzaldehyde derivatives are disubstituted gentisaldehyde (2,5-dihydroxybenzaldehyde) derivatives with a prenyl group at C-3 and a seven-carbon unbranched aliphatic chain at C-6 [35]. Bioassay-guided fractionation of *E. repens* leads to the isolation of two new benzaldehyde compounds, (*E*)-2-(hept-1-enyl)-3-(hydroxymethyl)-5-(3-methylbut-2-enyl)-benzene-1,4-diol (**42**; yellow solid) and (*E*)-4-(hept-1-enyl)-7-(3-methylbut-2-enyl)-2,3-dihydrobenzofuran-2,5-diol (**43**; yellow oil), along with five known benzaldehyde derivatives, including flavoglaucin, 2-(2′,3-epoxy-1′,3′-heptadienyl)-6-hydroxy-5-(3-methyl-2-butenyl)-benzaldehyde, auroglaucin, tetrahydroauroglaucin, and dihydroauroglaucin. Compounds **42** and **43** showed high structural similarities except that the carbinol group at C-7 in compound **42** was replaced by a hemiacetal group in compound **43** [37]. Gao et al. [19] also isolated flavoglaucin, 2-(2′,3-epoxy-1′,3′-heptadienyl)-6-hydroxy-5-(3-methyl-2-butenyl)-benzaldehyde, auroglaucin, tetrahydroauroglaucin, dihydroauroglaucin, and (*E*)-2-(hept-1-enyl)-3-(hydroxymethyl)-5-(3-methylbut-2-enyl)-benzene-1,4-diol from the fungus *E. repens*. 

Two new benzaldehyde derivatives named (3′*S**, 4′*R**)-6-(3′,5-epoxy-4′-hydroxy-1′-heptenyl)-2-hydroxy-3-(3″-methyl-2″-butenyl)-benzaldehyde (**44**; yellow oil) and 3′-OH-tetrahydroauroglaucin (**45**; yellow oil) were isolated from a gorgonian-derived *Eurotium* sp. These two compounds could non-enzymatically transform into pairs of enantiomers or epimers, respectively, with opposite configurations at C-3′; thus, they are possibly artifacts formed during the extraction/isolation process [40]. Two new benzaldehyde derivatives named cristaldehyde A (**46**; yellow powder) and cristaldehyde B (**47**; yellow powder) were isolated from the fungus *E. cristatum* in 2019. Compound **46** contains a dibenzannulated 6,6-spiroketal skeleton and is a racemic mixture of easily interconvertible enantiomers [38]. It is worth noting that six benzaldehyde derivatives, including flavoglaucin, isodihydroauroglaucin, 2-(2′,3-epoxy-1′,3′-heptadienyl)-6-hydroxy-5-(3-methyl-2-butenyl)-benzaldehyde, 2-(2′,3-epoxy-1′,3′,5′-heptatrienyl)-6-hydroxy-5-(3-methyl-2-butenyl)-benzaldehyde, tetrahydroauroglaucin, and dihydroauroglaucin, were discovered in Fuzhuan-brick-tea-derived *E. cristatum*. *E. cristatum* is the only dominant fungus in Fuzhuan brick tea, which is responsible for the colour, taste, and health benefits of Fuzhuan brick tea [72,73,74,75]. These benzaldehyde derivatives may have a major impact on the sensory quality and health benefits of Fuzhuan brick tea [14] (Figure 2).

### 2.3. Indole Diketopiperazine Alkaloids

Indole diketopiperazine alkaloids constitute a crucial class of important secondary metabolites, and they are widely distributed in filamentous fungi, especially in the genus *Eurotium* [18]. Indole diketopiperazine alkaloids are formed via the condensation of certain amino acids, including tryptophan, proline, and leucine [76]. Due to their significant biological activities, including antimicrobial, antiviral, anticancer, immunomodulatory, antioxidative, and insecticidal activities, indole diketopiperazine alkaloids in the genus *Eurotium* are attracting increasing attention [41,42].

Swine-rejected feed was found to have a high propagule density of *Eurotium* sp. Additionally, echinulin (**48**) was both detected in this feed and isolated from the *E. repens* derived from it [43]. Although significant differences in the metabolite composition were observed between the feed-derived and marine-derived *E. repens*, the biosynthesis of echinulin was conserved in *E. repens* regardless of its origin [24]. Kimoto et al. [48] isolated neoechinulin A (**49**) from marine fungus *E. rubrum* Hiji 025, and further synthesised this compound according to its natural configuration. Slack et al. [44] investigated the metabolites in *E. herbariorum*, *E. amstelodami*, and *E. rubrum*, which are common in the built environment of Canadian homes. Neoechinulin B (**50**) and neoechinulin A were the major metabolites, but preechinulin (**51**), neoechinulin E (**52**), and echinulin were the minor metabolites in *E. amstelodami* and *E. rubrum*. *E. herbariorum* also produced a small amount of neoechinulin E. In addition, a new spirocyclic diketopiperazine alkaloid, 7-*O*-methylvariecolortide A (**53**; yellow amorphous powder), was isolated from the mangrove plant *Hibiscus tiliaceus*-derived *E. rubrum*, along with variecolortides A-C (**54–56**). Structurally, compounds **53–56** represent the unique spiro-anthronopyranoid diketopiperazine skeleton with a stable hemiaminal functional group. Further, a hydroxyl group in compound **54** is replaced by a methoxyl group at C-7 in compound **53** [52]. Fructigenine A (**57**) bearing a reverse-prenyl group was isolated from *Eurotium* sp. SF-5130 [54].

A new diketopiperazine dimer, namely eurocristatine (**58**; white crystals), was isolated and identified from *E. cristatum*, along with previously reported dioxopiperazine alkaloids including variecolorin J (**59**), echinulin, neoechinulin A, and neoechinulin E [26]. The semi-mangrove plant *Hibiscus tiliaceus*-derived *E. rubrum* was cultivated by Yan et al. [28], and one new dioxopiperazine alkaloid, 12-demethyl-12-oxo-eurotechinulin B (**60**; colourless amorphous powder), was further isolated from this fungal strain, along with six known compounds, including variecolorin J, variecolorin G (**61**), eurotechinulin B (**62**), cryptoe-chinuline G (**63**), alkaloid E-7 (**64**), and isoechinulin B (**65**). The structures of compounds **60** and **62** are similar, except that the Me-C (**12**) of compound **62** is replaced by a C (12) =O group in compound **60**. Du et al. [41] also found four new alkaloids named cristatumins A-D (**66–69**) in the culture extract of *E. cristatum* EN-220, along with six known congeners including isoechinulin A (**70**), tardioxopiperazine A (**71**), echinulin, neoechinulin A, preechinulin, and variecolorin G. This is the first report that the alanine residue in the 2,5-diketopiperazine moiety of compound **49** is replaced by the serine residue in compound **66**. The C-20 Me group in compound **48** is replaced by a CH_2_OH group in compound **67**. Compound **68** is an almost symmetrical molecule consisting of two indole diketopiperazine moieties. Compound **69** is a ring-opened diketopiperazine derivative of compound **52**. Equally, a pyrrolidinoindoline diketopiperazine alkaloid named cristatumin E (**72**; yellow amorphous powder) was isolated from the alga-derived *E. herbariorum* HT-2 [55]. 

In 2018, three new indole diketopiperazine alkaloids of isoechinulin type named rubrumazines A-C (**73–75**) and 13 related analogues were isolated and identified from *E. rubrum* MA-150, a fungus obtained from mangrove-derived rhizospheric soil collected from the Andaman Sea coastline in Thailand. These 13 related analogues were dehydroechinulin (**76**), variecolorin E (**77**), dihydroxyisoechinulin A (**78**), variecolorin L (**79**), tardioxopiperazine (**80**), _L_-alanyl-_L_-tryptophan anhydride (**81**), echinulin, neoechinulin A, neoechinulin E, variecolortide B, variecolortide C, variecolorin G, and isoechinulin A. Compounds **73–75** possess an oxygenated prenyl group either at C-7 (**73** and **74**) or at C-5 (**75**) [49]. A new prenylated indole diketopiperazine alkaloid named cristatumin F (**82**; colourless powder) was isolated from the Fuzhuan-brick-tea-derived *E. cristatum*, along with four known compounds, including variecolorin O (**83**), echinulin, neoechinulin A, and dehydroechinulin. Structurally, compound **82** is a diketopiperazine congener to compound **48**. An alanine unit in compound **48** is replaced by a valine unit in the 2,5-diketopeperazine moiety in compound **82** [42]. Four new indole diketopiperazine derivatives (**84–87**) and nine known congeners (**88–91**, **48**, **50**, **64**, **74**, **76**) were identified from a culture extract of *E. cristatum* EN-220. Compounds **84–91** were *N*-(4′-hydroxyprenyl)-cyclo(alanyltryptophyl) (**84**), isovariecolorin I (**85**), 30-hydroxyechinulin (**86**), 29-hydroxyechinulin (**87**), rubrumline M (**88**), neoechinulin C (**89**), didehydroechinulin (**90**), and variecolorin H (**91**) [45]. In addition, (11*R*,14*S*)-3-(1*H*-indol-3ylmethyl)6-isopropyl-2,5-piperazinedione (**92**) was isolated from the culture of *E. chevalieri* KUFA 0006 [20].

Zhong et al. [56] isolated three pairs of spirocyclic diketopiperazine enantiomers named variecolortins A-C (**93–95**) from marine-derived fungus *Eurotium* sp. SCSIO F452. Compound **93** possesses an unprecedented highly functionalised benzo[*f*]pyrazino [2,1*-b*] [1,3]oxazepine new carbon skeleton comprising a 2-oxa-7-azabicyclo[3.2.1]octane core. Compounds **94–95** represent rare examples of a 6/6/6/6 tetracyclic cyclohexene-anthrone carbon scaffold. Further, Zhong et al. [46] isolated and characterised three new prenylated indole 2,5-diketopiperazine alkaloids named eurotiumins A-C (**96–98**; white crystals, white solid, and yellow oil, respectively) from the *Eurotium* sp. SCSIO F452 in the same year. Compounds **96** and **97** are a pair of diastereomers presenting a hexahydropyrrolo[2,3-*b*]indole skeleton. The structures of compounds **96** and **97** are assigned as *2S,3R,9S,12S-cyclo*-2-dimethylallyl-3-hydroxy-_L_-Trp-_L_-Ala and *2R,3S,9S,12S-cyclo*-2-dimethylallyl-3-hydroxy-_L_-Trp-_L_-Ala, respectively. The structures of compounds **98** and **50** are similar, except that an olefinic methylene in compound **50** is transformed into an olefinic methine substituted by a doublet methyl in compound **98**. In 2021, Elsebai et al. [57] found a diketopiperazine indole alkaloid named fintiamin (**99**) in a marine sponge *Ircinia variabilis*-derived *Eurotium* sp. Compound **99** is a lipophilic terpenoid-dipeptide hybrid molecule, sharing similar synthetic pathways to compound **48** (Figure 3).

### 2.4. Other Compounds

Six meroterpenoid-type terpenoids named chevalones A-D (**100–103**; colourless crystals, colourless crystals, white solid, and white solid, respectively) and aszonapyrones A-B (**104–105**) and a terpenoid pyrrolobenzoxazine named CJ-12662 (**106**) have been isolated from E. chevalieri [58]. There were 11 steroids isolated from *E. rubrum*: 3*β*,5*α*-dihydroxy-10*α*-methyl-6*β*-acetoxy-ergosta-7,22-diene (**107**; colourless crystals), 3*β*,5*α*-dihydroxy-6*β*-acetoxyergosta-7,22-diene (**108**), (22*E*,24*R*)-ergosta-7,22-dien-3*β*-ol (**109**), (22*E*,24*R*)-ergosta-7,22-dien-6*β*-methoxy-3*β*,5*α*-diol (**110**), (22*E*,24*R*)-ergosta-7,22-dien-3*β*,5*α*,6*β*-triol (**111**), (22*E*,24*R*)-ergosta-7,22-dien-3*β*,5*α*,6*α*-triol (**112**), (22*E*,24*R*)-3*β*,5*α*,9*α*-trihydroxyergosta-7,22-dien-6-one (**113**), (22*E*,24*R*)-3*β*,5*α*-dihydroxyergosta-7,22-dien-6-one (**114**), (22*E*,24*R*)-5*α*,8*α*-epidioxyergosta-6,22-dien-3*β*-ol (**115**), (22*E*,24*R*)-5*α*,8*α*-epidioxyergosta-6,22-dien-3*β*-acetate (**116**), and (22*E*,24*R*)-ergosta-4,6,8(14),22-tetraen-3-one (**117**) [59]. There were 13 salicylaldehyde derivatives, including euroticins A-I (**118–126**), salicylaldehydiums A-B (**127–128**), and asperglaucins A-B (**129–130**) isolated from *Eurotium* sp. SCSIO F452 [15,34,60,61] or *E. chevalieri* SQ-8 [17]. In addition, eight mycotoxins were isolated from Eurotium species contain citrinin (**131**), ochratoxin A (**132**), gliotoxin (**133**), aflatoxins (**134**), and sterigmatocystin (**135**) from the *Eurotium* group [62]; a benzodiazepine-type mycotoxin cyclopenol (**136**) from *Eurotium* sp. SF-5130 [54]; and mycophenolic acid (**137**) from *E. repens* [63]. Three indole alkaloids, 2-(2-methyl-3-en-2-yl)-1*H*-indole-3-carbaldehyde (**138**), and (2,2-dimethylcyclopropyl)-1*H*-indole-3-carbaldehyde (**139**) were isolated from *E. chevalieri* KUFA 0006 [20], and 2-(1,1-dimethyl-2-propen-1-yl)-1*H*-indole-3-carboxaldehyde (**140**) was isolated from *Eurotium* sp. SCSIO F452 [64].

Other compounds isolated from *Eurotium* species include ergosterol (**141**) [58], 2[(2,2-dimethylbut-3-enoyl)amino]benzoic acid (**142**; yellow viscous liquid), 6,8-dihydroxy-3-(2-hydroxypropyl)-7-methyl-1*H*-isochromen-1-one (**143**; yellow viscous liquid), palmitic acid, ergosterol 5,8-endoperoxide (**144**) [20], (11*S*,14*R*)-cyclo(tryptophylvalyl) (**145**; white crystal), cinnalutein (**146**), *cyclo*-_L_-Trp-_L_-Ala (**147**) [25], eurochevalierine (**148**; yellow needles), and sequiterpene (**149**) [58] from *E. chevalieri*; zinniol (**150**), butyrolactone I (**151**), aspernolide D (**152**), vermistatin (**153**), methoxyvermistatin (**154**), eurothiocin A (**155**; colourless oil), eurothiocin B (**156**; white amorphous solid) [65], and 7-isopentenylcryptoechinuline D (**157**) [28] from *E. rubrum*; methyl linoleate (**158**; yellow oil) [64], *cyclo*-(_L_-Pro-_L_-Phe) (**159**) [46], eurotinoids A-C (**160–162**), dihydrocryptoechinulin D (**163**) [66], and (±)-Eurotone A (**164**) [67] from *Eurotium* sp. SCSIO F452; 5,7-dihydroxy-4-methylphthalide (**165**) from *E. repens* [37]; cristatumside A (**166**) from *E. cristatum* EN-220 [31]; (±)-eurotiumides A-G (**167-173**) from *Eurotium* sp. XS-200900E6 [21]; alkaloid viridicatol (**174**) from *Eurotium* sp. SF-5130 [54]; a β-hydroxy acid named monacolin K (**175**) [68] and a quinone derivative, cristaquinone A (**176**) [38], from *E. cristatum*; and a glycoside isotorachrysone 6-*O*-*α*-_D_–ribofuranoside (**177**) from *E. cristatum* EN-220 [31] (Figure 4).

## 3. Bioactivities of Secondary Metabolites from *Eurotium*

Pharmacological investigations have affirmed that the structurally distinctive compounds extracted from *Eurotium* species exhibit a spectrum of biological activities, encompassing antioxidative, antimicrobial, cytotoxic, antitumor, insecticidal, antimalarial, and anti-inflammatory properties. We provide a review of these functional secondary metabolites to provide a scientific basis for the development of functional foods using *Eurotium* as a fermentative strain.

### 3.1. Antioxidative Activity

Numerous studies have demonstrated the exceptional antioxidative activity of metabolites isolated from *Eurotium* species. Further, the absolute and stereoscopic configurations affect the antioxidative activity of these compounds [46,56]. Ishikawa et al. [77] discovered that flavoglaucin (**31**) was an excellent antioxidant and synergist with tocopherol. The antioxidative and synergistic effects of flavoglaucin and its derivatives largely depend on their hydroxy group, which does not form hydrogen bonds with the formyl group in the molecule. These compounds are found in a variety of foods fermented by *Eurotium* and contribute to their functional activity [78]. Li et al. [51] assessed the antioxidative activity of metabolites isolated from a marine mangrove plant-derived endophytic fungus *E. rubrum* using the 1,1-diphenyl-2-picrylhydrazyl (DPPH) radical scavenging assay. They found that neoechinulin E (**52**) showed a strong radical scavenging activity with half maximal inhibitory concentration (IC_50_) values of 46.0 μM, which were stronger than that of the well-known synthetic antioxidant butylated hydroxytoluene (IC_50_ = 82.6 μM). Eurorubrin (**19**) and 2-*O*-methyleurotinone (**10**) also displayed strong radical scavenging activity with IC_50_ values of 44.0 and 74.0 μM, respectively, while 2-*O*-methyl-4-*O*-(*α*-_D_-ribofuranosyl)-9-dehydroxyeurotinone (**17**), 3-*O*-(*α*-_D_-ribofuranosyl)-questin (**20**), 2-*O*-methyl-9-dehydroxyeurotinone (**18**), asperflavin (**15**), and questin (**1**) only showed weak or moderate activity [23]. In 2009, a study by Miyake et al. [35] demonstrated that isodihydroauroglaucin (**34**), auroglaucin (**39**), dihydroauroglaucin (**41**), tetrahydroauroglaucin (**40**), and flavoglaucin exhibited the high radical scavenging capacities of DPPH and superoxide when compared to *α*-tocopherol (a standard antioxidant for the scavenging capacity). The structures of 1′-monoene or 1′,3′-diene in the substituent formed by the seven-carbon aliphatic chain of dihydroauroglaucin and tetrahydroauroglaucin may be related to their high radical scavenging activity. Subsequently, Miyake et al. [16] found that isoechinulin A (**70**) exhibited higher radical scavenging activity than *α*-tocopherol. Asperflavin, isoechinulin B, neoechinulin B (**50**), and variecolorin O (**83**) were found to have a similar activity to *α*-tocopherol in respect to DPPH radical scavenging.

The compounds eurotiumin C (**98**), dehydroechinulin (**76**), variecolorin G (**61**), isoechinulin A, variecolorin O, neoechinulin B, and echinulin (**48**) showed significant radical scavenging activity against DPPH with IC_50_ values of 13, 19, 4, 3, 24, 13, and 18 μM, respectively. These values were comparable or superior to that of ascorbic acid (Vc) (IC_50_ = 23 μM). Further, the diprenylated analogs (compounds **61** and **70**) were found to have higher radical scavenging activity than the monoprenylated ones (compounds **96–98**, **83**, and **50**) and triprenylated ones (compounds **76** and **48**). The absolute configurations of the C-2 and C-3 in eurotiumin A (**96**) and B (**97**) may affect their radical scavenging activity [46]. (+)-variecolortin A (**93**) showed radical scavenging activity against DPPH with an IC_50_ value of 58.4 μM, while the IC_50_ value of (-)-variecolortin A (**93**) was 159.2 μM. This implied that the stereoscopic configuration affects the biological activities of these two compounds [56]. In addition, the compounds (±)-eurotinoids A-C (**160–162**) and dihydrocryptoechinulin D (**163**) showed significant antioxidative activity against DPPH with IC_50_ values ranging from 3.7 to 24.9 μM, which were more potent than that of the positive control Vc [66]. The compounds (+)-euroticins B and (-)-euroticins B (**119**) showed remarkable DPPH radical scavenging activity with a concentration of 50% leading to maximal effect (EC_50_) values of 37.5 and 21.6 μM, which were superior or comparable to that of the positive control Vc (EC_50_ = 27.9 μM) [15]. In 2021, Zhong et al. [34] found that (+)-euroticin C and (-)-euroticin C (**120**) showed significant DPPH radical scavenging activity with EC_50_ values of 27.00 and 30.27 μM [60], but (±)-euroticin F (**123**) and G (**124**) showed weak activity, with EC_50_ values ranging from 41.40 to 77.07 μM. In addition, the compound neoechinulin A (**49**) showed antioxidative activity against peroxynitrite derived from SIN-1 in neuronal PC12 cells [48]. Nonetheless, the antioxidative activity of the metabolites isolated from *Eurotium* species was mainly measured using in vitro experiments, so in vivo tests in animal models should be encouraged.

### 3.2. Antimicrobial Activity

Microbial interference poses a significant threat to human health, and the search for antimicrobial compounds from *Eurotium* species represents a promising strategy to combat the escalating challenges posed by human and plant pathogens, particularly drug-resistant strains. Further, the antimicrobial activity of *Eurotium* species may be related to anthraquinones [79,80,81]. As early as 1980, erythroglaucin (**3**) was found to have slight antibacterial activity against *Bacillus brevis*, *Bacillus subtilis*, and *Streptomyces viridochromogenes*. However, rubrocristin (**6**) and physcion (**2**) had no significant antimicrobial activity, indicating that the number and location of the hydroxyl groups might play an important role in the antibacterial activity of polyhydroxyanthraquinones [22]. Chevalone C (**102**), eurochevalierine (**148**), and CJ-12662 (**106**) demonstrated antimycobacterial activity against *Mycobacterium tuberculosis* with minimal inhibitory concentration (MIC) values of 6.3, 50.0, and 12.5 μg/mL, respectively [58]. In 2012, Du et al. [41] evaluated the antimicrobial activities of compounds isolated from *E. cristatum* against two bacteria (*Staphylococcus aureus* and *Escherichia coli*) and five plant-pathogenic fungi (*Valsa mali*, *Sclerotinia miyabeana*, *Alternaria brassicae*, *Physalospora obtuse*, and *Alternaria solania*). The MIC value of the positive control chloramphenicol against *E. coli* and *S. aureus* was 4 μg/mL. Cristatumin A (**66**) and tardioxopiperazine A (**71**) displayed potent inhibitory activity against *E. coli* and *S. aureus* with MIC values of 64 and 8 μg/mL, whereas cristatumin D (**69**) and echinulin showed weak activity against *S. aureus*, each creating an inhibition zone of 8 mm at 100 μg/disk (the MICs were not determined). In addition, the compound 9-dehydroxyeurotinone (**23**) isolated from *E. rubrum* showed weak antibacterial activity against *E. coli* with an inhibition zone of 7.0 mm at 100 μg/disk, while amphotericin B had an inhibition zone of 11.0 mm at 20 μg/disk as the control [28].

Gao et al. [19] evaluated the antimicrobial activities of isolated metabolites from *E. repens* against five bacteria (*S. aureus*, *methicillin-resistant S. aureus*, *P. aeruginosa*, *M. intracellulare*, and *E. coli*) and five pathogenic fungi (*Candida. albicans*, *Candida glabrata*, *Candida krusei*, *Cryptococcus neoformans*, and *Aspergillus fumigatus*). Flavoglaucin, tetrahydroauroglaucin, and 2-(2′,3-epoxy-1′,3′-heptadienyl)-6-hydroxy-5-(3-methyl-2-butenyl)-benzaldehyde (**35**) exhibited antibacterial activity against *S. aureus* with IC_50_ values of 14.32, 13.51, and 7.75 μg/mL, respectively; (*E*)-2-(hept-1-enyl)-3-(hydroxymethyl)-5-(3-methylbut-2-enyl)-benzene-1,4-diol (**42**) and compounds **31** and **35** were active against *S. aureus* with IC_50_ values of 11.97, 10.41, and 5.40 μg/mL, respectively; auroglaucin, dihydroauroglaucin, and compounds **35**, **40**, and **42** showed antifungal activity against *C. glabrata* with IC_50_ values of 7.33, 2.39, 1.13, 6.15, and 7.17 μg/mL, respectively. Compound **35** and 5,7-dihydroxy-4-methylphthalide (**165**) showed antifungal activity against *C. neoformans* with IC_50_ values of 5.31 and 18.08 μg/mL, respectively; only auroglaucin exhibited moderate antifungal activity against *C. krusei* with an IC_50_ value of 10.93 μg/mL. In addition, cristatumin E (**72**) showed weak antibacterial activity against *E. aerogenes* and *E. coli* with IC_50_ and MIC values of 8.3, 44.0, and 44.0 μM, respectively [55]. The compounds 3-*O*-(*α*-_D_-ribofuranosyl)-questinol (**21**) and eurorubrin showed weak inhibitory activity against *E. coli* with MIC values of 32 and 64 μg/mL, while chloramphenicol had an MIC value of 4 μg/mL as control [31]. Emodin (**4**) not only showed moderate antibacterial activity against the Gram-positive bacteria but also exhibited a strong synergistic association with oxacillin against methicillin-resistant *S. aureus* (MRSA) [20]. In 2019, asperflavin was found to be active against *S. aureus* (MIC of 64 μg/mL) and *S. pneumoniae* Monza-82 (MIC of 32 μg/mL). Dihydroauroglaucin was active against the Gram-positive bacteria with MIC values of 128 μg/mL, 64 μg/mL, and 8 μg/mL on *S. aureus*, *E. faecalis*, and *S. pneumoniae*, respectively. Compound **41** was previously considered inactive against reference and MRSA *S. aureus* strains [25]. Neoechinulin A, _L_-alanyl-_L_-tryptophan anhydride (**81**), dihydroxyisoechinulin A (**78**), and questin showed obvious antibacterial activity against *B. cereus* and *P. vulgaris* with MIC values of 1.56 to 25 μM when ciprofloxacin (MIC values of 0.78 and 0.20 μM, respectively) was used as the positive control and DMSO (25 μM) was used as the negative control [11]. Asperglaucins A (**129**) and B (**130**) exhibited potent antibacterial activities against *Pseudomonas syringae* pv. *actinidae* and *B. cereus*, with all having MIC values of 6.25 μM. Compound **129** also exhibited a weak inhibitory effect against MRSA with an MIC value of 25 μM. The activity of compounds **129** and **130** is probably due to their heterocyclic fraction [17]. Notably, the above intriguing new compounds, which exhibit excellent antimicrobial properties, could be used as the leading compounds for the development of new drugs in the future.

### 3.3. Cytotoxicity and Antitumour Activities

The cytotoxicity and antitumor activities of *Eurotium* species have been extensively studied since the 1970s. Podojil et al. [4] reported that physcion had cytotoxicity towards HeLa cells with an IC_50_ value of 0.1 μg/mL. Smetanina et al. [24] found that physcion, asperflavin, and tetrahydroauroglaucin exhibited cytotoxic activity against the sex cells of sea urchin *Strongylocentrotus intermedius* at concentrations of 25 μg/mL, 10 μg/mL, and 0.5 μg/mL, respectively. In addition, compounds chevalone C, chevalone D, eurochevalierine, and CJ-12662 had respective IC_50_ values against BC1 human breast cancer cells of 8.7, 7.8, 5.9, and 7.6 μg/mL, respectively. Compounds chevalone B (**101**) and eurochevalierine exhibited cytotoxicity against KB human epidermoid carcinoma cells and NCI-H187 small cell lung cancer cells with IC_50_ values in the range of 2.9 to 9.8 μg/mL [58]. In 2012, Yan et al. [28] investigated the cytotoxic activities of some *E. rubrum*-derived alkaloids and anthraquinones against seven tumor cell lines, including MCF-7, SW1990, SMMC-7721, Hela, HepG2, NCI-H460, and Du145. 9-dehydroxyeurotinone exhibited cytotoxic activity with an IC_50_ value of 25 μg/mL against SW1990; variecolorin G exhibited cytotoxic activity with IC_50_ values of 20, 22, and 20 μg/mL against HepG2, NCI-H460, and Hela, respectively; alkaloid E-7 (**64**) exhibited cytotoxic activity with IC_50_ values of 20, 20, 20, and 30 μg/mL against MCF-7, SW1990, SMMC-7721, and Hela cells, respectively; 12-demethyl-12-oxo-eurotechinulin B (**60**) exhibited slight cytotoxic activity with an IC_50_ value of 30 μg/mL against SMMC-7721, and only emodin exhibited moderate cytotoxic activity with an IC_50_ value of 15 μg/mL against Du145. Besides, cristatumin E showed cytotoxicity against the K562 tumor cell line with an IC_50_ value of 8.3 mM [55].

Rubrumol (**25**) showed relaxation activity for topoisomerase I, with an IC_50_ value of 23 μM [27]. In 2018, Zhong et al. [56] found that (+)-variecolortin B (**94**) showed moderate cytotoxicity against the SF-268 and HepG2 cell lines with IC_50_ values of 12.5 and 15.0 μM, while (+)-variecolortin C (**95**) had the values of 30.1 and 37.3 μM. Compounds (-)-variecolortin B and (-)-variecolortin C were inactive (>100 μM) for SF-268 and HepG2 cells. In addition, compound (+)-dihydrocryptoechinulin D showed moderate cytotoxicity against the SF-268 and HepG2 cell lines with IC_50_ values of 51.7 and 49.9 μM, and (-)-dihydrocryptoechinulin D had values of 97.3 and 98.7 μM, respectively. Thus, (+)-enantiomers exhibited more valid activities than the corresponding (-)-enantiomers [66]. Flavoglaucin displayed weak cytotoxic activity against HepG2 and HeLa with IC_50_ values of 41.48 and 33.60 μM, respectively [38]. (-)-Salicylaldehydium A (**127**) showed cytotoxic activity against SF-268 and HepG2 cells with IC_50_ values of 91.0 and 95.5 μM, respectively [61]. (±)-Euroticin F, (±)-euroticin I (**126**), and (±)-eurotirumin (**29**) exhibited moderate cytotoxic activity with IC_50_ values ranging from 12.74 to 55.5 μM [34]. Euroticin C exerted moderate cytotoxic activity against human SF-268, MCF-7, HepG-2, and A549 cells [60]. However, the compounds’ relative toxicities are unknown; few research works on target organ toxicities or even side effects exist in the report.

### 3.4. Insecticidal Activity

Brine shrimp (*Artemia salina*), known for their high sensitivity to toxins and ease of cultivation, serve as a model organism frequently employed by researchers for screening substances with insecticidal activity [45,49]. In 2012, Du et al. [41] reported that cristatumin B (**67**), isoechinulin A, and variecolorin G exhibited moderate lethal activity against brine shrimp with median lethal dose (LD_50_) values of 74.4, 16.9, and 42.6 μg/mL, respectively. The structure–activity relationships indicated that the number and substituted position of the isoprenic chains are important for the insecticidal activities of these compounds. As for lethality against brine shrimp, eurorubrin exhibited moderate activity with a lethal rate of 41.4% at a concentration of 10 μg/mL [31]. Rubrumazine B (**74**), dehydroechinulin, and neoechinulin E exhibited potent activity against brine shrimp with LD_50_ values of 2.43, 3.53, and 3.93 μM, respectively, which were lower than that of the positive control colchicine (LD_50_ 19.4 μM) [49]. In addition, Du et al. [45] showed that isovariecolorin I (**85**), neoechinulin C (**89**), alkaloid E-7, and didehydroechinulin (**90**) displayed potent activity against brine shrimp with ] LD_50_ values of 19.4, 70.1, 19.8, and 27.1 μg/mL, respectively.

Some *Eurotium*-derived compounds were evaluated for their antifouling activities against the larval settlement of the barnacle *Balanus amphitrite*, which is one of the representative marine fouling organisms. Compounds (±)-eurotiumides A-D (**167–170**) inhibited the barnacle larval settlement with EC_50_ values < 25.0 μg/mL, which was lower than the standard requirement established by the U.S. Navy. Specifically, (+)-eurotiumide B, (-)-eurotiumide B, (+)-eurotiumide D, and (-)-eurotiumide D with cis configurations of H-3/H-4 exhibited better antifouling activities (EC_50_ values of 1.5, 0.7, 2.3, and 1.9 μg/mL) than the corresponding (+)-eurotiumide A, (-)-eurotiumide A, (+)-eurotiumide C, and (-)-eurotiumide C (trans configurations of H-3/H-4; EC_50_ values of 19.4, 22.5, 20.2, and 23.2 μg/mL). This suggested that the relative configuration of H-3/H-4 might be an important factor affecting antifouling activity [21]. In addition, the compounds neoechinulin A and echinulin inhibited the barnacle larval settlement with EC_50_ values of 15.0 and 17.5 μg/mL, respectively [47].

### 3.5. Antimalarial Activity

In 2012, Gao et al. [19] measured the antiprotozoal activity of secondary metabolites from the fungus *E. repens* in vitro against chloroquine-sensitive and chloroquine-resistant strains of *Plasmodium falciparum*. The compounds flavoglaucin, 2-(2′,3-epoxy-1′,3′-heptadienyl)-6-hydroxy-5-(3-methyl-2-butenyl)-benzaldehyde, auroglaucin, tetrahydroauroglaucin, and (*E*)-2-(hept-1-enyl)-3-(hydroxymethyl)-5-(3-methylbut-2-enyl)-benzene-1,4-diol exhibited moderate antimalarial activities with IC_50_ values in the range of 1.1–3.0 μg/mL, among which compound **39** displayed the highest antimalarial activity. This suggested the three consecutive double bonds in compound **39** might contribute to the enhancement of antimalarial activity. In addition, chevalone D, eurochevalierine, and CJ-12662 exhibited antimalarial activity against *Plasmodium falciparum* with IC_50_ values of 3.1, 3.4, and 6.5 μg/mL, respectively [58].

### 3.6. Anti-Inflammatory Activity

Kim et al. [50] demonstrated that neoechinulin A had an anti-inflammatory effect on lipopolysaccharide-stimulated RAW264.7 macrophages. Further, compound **49** blocked the activation of nuclear factor-kappa B (NF-κB) by inhibiting the phosphorylation and degradation of inhibitor kappa B-*α*, and decreased p38 mitogen-activated protein kinase (MAPK) phosphorylation. The anti-inflammatory effect of compound **49** was thus attributed to the inhibition of the NF-κB and p38 MAPK pathways. In addition, the compounds flavoglaucin, isotetrahydroauroglaucin (**33**), and asperflavin were found to inhibit the production of pro-inflammatory mediators and cytokines, including tumor necrosis factor-*α*, interleukin-1β, interleukin-6, nitric oxide (NO), prostaglandin E2, nitric oxide synthase, and cyclooxygenase-2 [30,32,36]. Cristaldehyde A (**46**) and cristaquinone A (**176**) inhibited the NO production in lipopolysaccharide-induced RAW264.7 cells, with IC_50_ values of 12.26 and 1.48 μM when paclitaxel was used as a positive control, with an IC_50_ value of 41.00 μM [38].

### 3.7. Other Activities

Several isolated compounds have certain unique biological activities, including a good binding affinity for human opioid or cannabinoid receptor activity, inhibiting protein tyrosine phosphatase 1B activity, alleviating insulin resistance activity, inhibiting caspase-3 activity, inhibiting *α*-glucosidase activity, and antiviral activity.

The compounds flavoglaucin, auroglaucin, tetrahydroauroglaucin, (*E*)-2-(hept-1-enyl)-3-(hydroxymethyl)-5-(3-methylbut-2-enyl)-benzene-1,4-diol, and (*E*)-4-(hept-1-enyl)-7-(3-methylbut-2-enyl)-2,3-dihydrobenzofuran-2,5-diol showed a good binding affinity for human opioid or cannabinoid receptors. This finding may contribute to the discovery of new selective ligands for opioid or cannabinoid receptors [37]. Fructigenine A (**57**), viridicatol (**174**), echinulin, flavoglaucin, and cyclopenol (**136**) were found to inhibit protein tyrosine phosphatase 1B activity with IC_50_ values of 10.7, 64.0, 29.4, 13.4, and 30.0 μM, respectively. This indicated that these compounds had potential for the treatment of type 2 diabetes and obesity [54]. In addition, eurocristatine (**58**) alleviated insulin resistance by increasing glucose consumption, glucose uptake, and glycogen content in high-glucose-induced HepG2 cells *in vitro*. Further, compound **58** improved glucose metabolism and alleviated insulin resistance in db/db diabetic mice by activating the phosphatidylinositol 3-kinase/protein kinase B signaling pathway [82].

The compounds 7-*O*-methylvariecolortide A (**53**), variecolortide B (**55**), and variecolortide C (**56**) showed an inhibitory effect on caspase-3 *in vitro*, with IC_50_ values of 1.7, 0.8, and 15.7 μM, respectively, when Ac-DEVD-CHO was used as a positive control (IC_50_ = 13.7 μM) [53]. Secondary metabolites isolated from the fungus *E. rubrum* SH-823 were examined for their *α*-glucosidase inhibitory activity. Eurothiocin A (**155**) and eurothiocin B (**156**) showed potent inhibitory potential (IC_50_ of 17.1 and 42.6 μM, respectively). Further, compounds **155** and **156** were competitive inhibitors of *α*-glucosidase [65]. In addition, compounds (±)-euroticin H (**125**) and (+)-euroticin G (**124**) exhibited significant inhibition against *α*-glucosidase with IC_50_ values of 16.31 and 38.04, which are even better than that of the positive control acarbose (IC_50_ of 32.92 μM) [34]. It is worth mentioning that significant antiviral activity for physcion and dihydroauroglaucin was discovered against two important human viral pathogens (herpes simplex virus 1 and influenza A virus) [25] (Figure 5).

## 4. Conclusions

*Eurotium*, a crucial genus within the *Aspergillus* family, has emerged as a significant source of bioactive compounds. Several factors contribute to its importance, including its widespread distribution, its role as a key microorganism in the fermentation of traditional foods and beverages (e.g., Fuzhuan brick tea), and its abundant production of secondary metabolites with promising bioactivities. Approximately 180 chemical components have been isolated from *Eurotium* species, spanning anthraquinoes, benzaldehyde derivatives, indol diketopiperazine alkaloids, and some other compounds. Various pharmacological activities, including antioxidative, antimicrobial, cytotoxic, antitumor, insecticidal, antimalarial, and anti-inflammatory activities, have been demonstrated in *Eurotium* species using numerous test models. However, further research employing in vivo models is imperative. In addition, secondary metabolites with health benefits should be introduced into the food industry to develop new functional foods. Most of the research has focused on three *Eurotium* species—*E. amstelodami*, *E. cristatum*, and *E. repens*—and should be further expanded to discover other species in the genus *Eurotium* from natural environments, such as the sea, with a view to introducing new strains for food fermentation. The other species in genus *Eurotium* the should be further studied, and this study will also provide information on the taxonomic relationships between *Eurotium* species. In addition, more attention should focus on the discovery of new secondary metabolites and their biological activities from fermented food/drink-derived and marine-derived *Eurotium* species. Delving into the pathways responsible for the formation of these metabolites is equally crucial for advancing our understanding of their potential applications.

## Figures and Tables

**Figure 1 foods-12-04452-f001:**
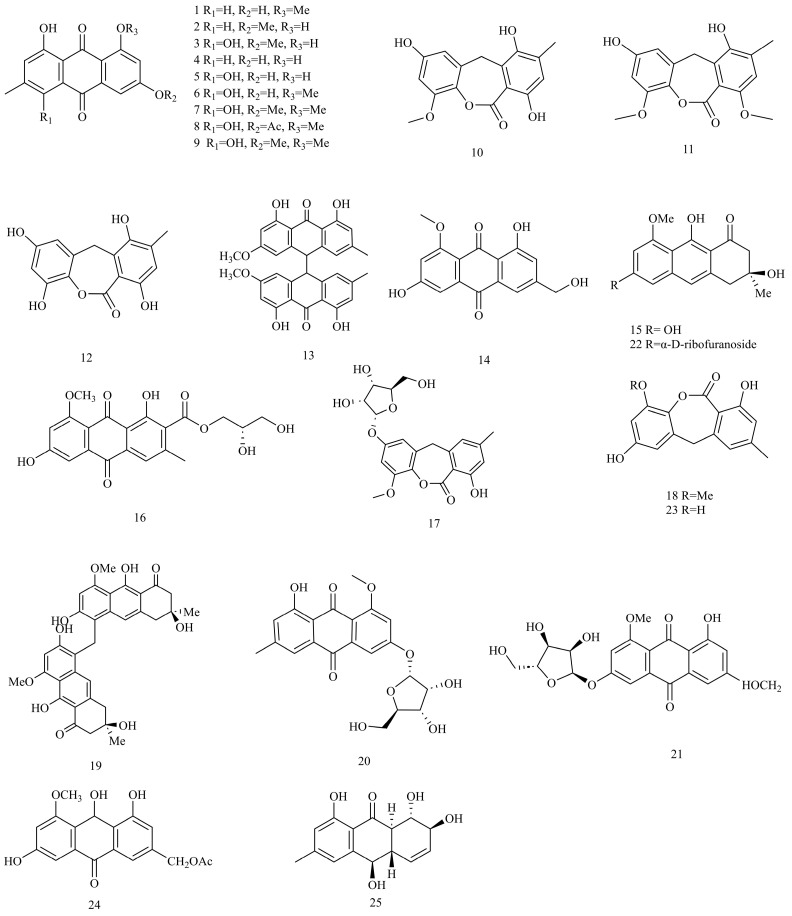
Structures of anthraquinones (compounds **1**–**25**).

**Figure 2 foods-12-04452-f002:**
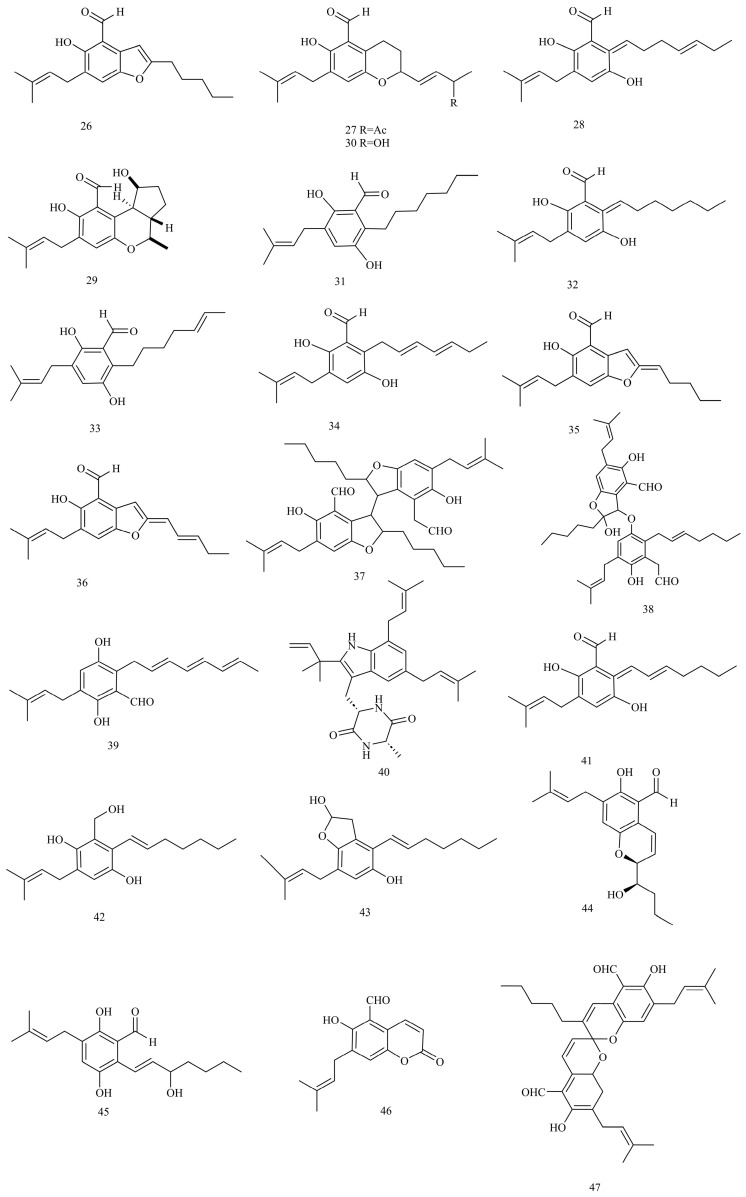
Structures of benzaldehyde derivatives (compounds **26**–**47**).

**Figure 3 foods-12-04452-f003:**
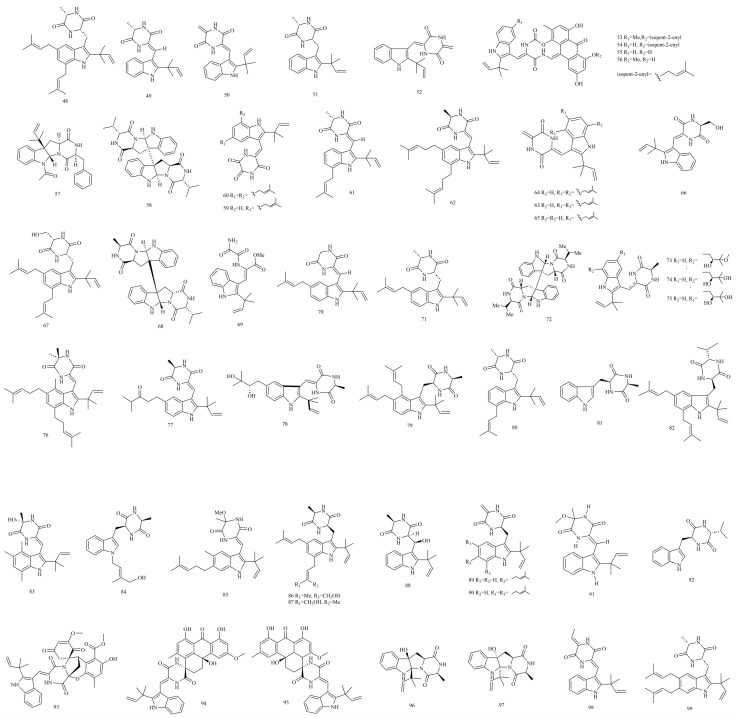
Structures of indole diketopiperazine alkaloids (compounds **48**–**99**).

**Figure 4 foods-12-04452-f004:**
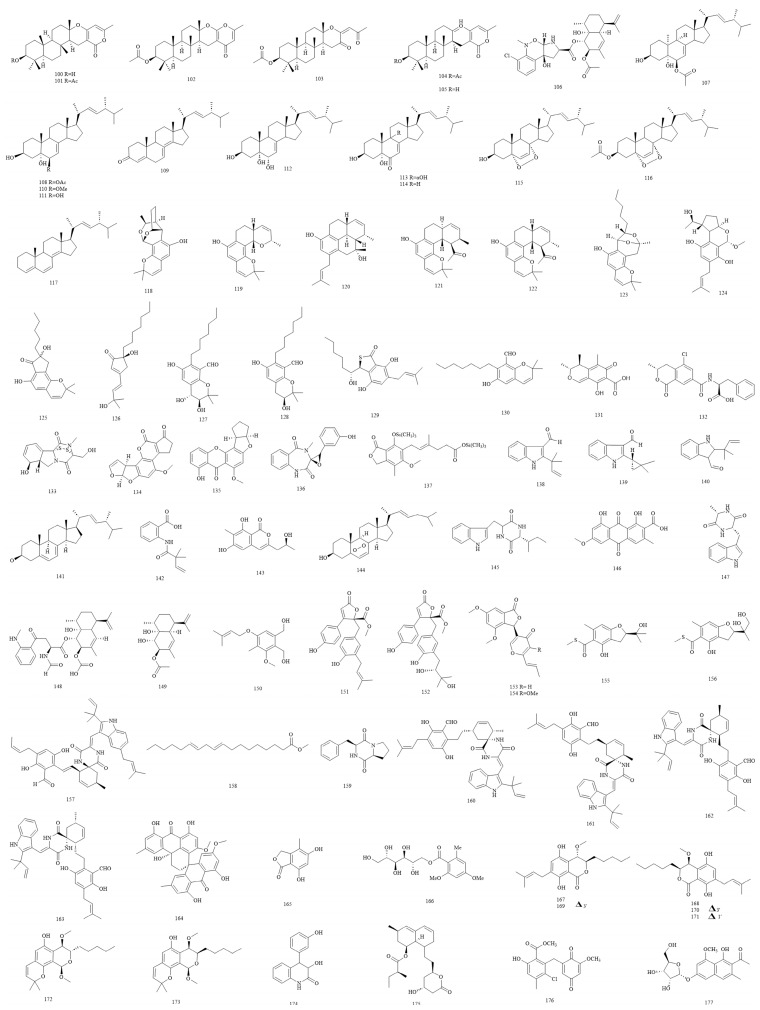
Structures of other compounds (compounds **100**–**177**).

**Figure 5 foods-12-04452-f005:**
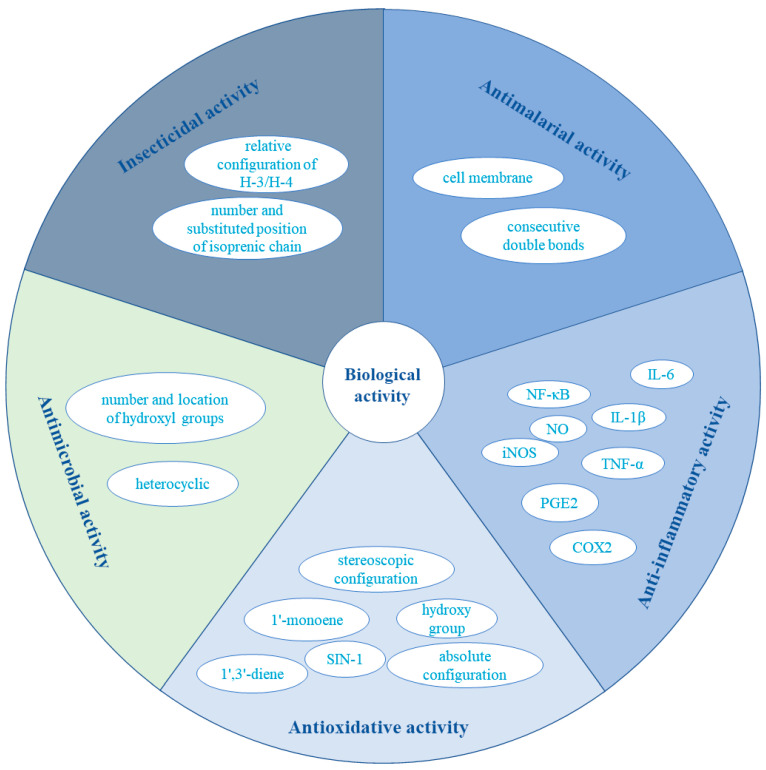
Overview of main biological activities.

**Table 1 foods-12-04452-t001:** Secondary metabolites from the genus *Eurotium* and their biological activities.

NO.	Compound Class and Name	Bioactivity	Source	Ref.
Anthraquinones				
1	questin	antimicrobial activity	*Eurotium* sp. M30 XS-2012	[11]
			*E. herbariorum* NU-2	[16]
			*E. chevalieri* KUFA 0006	[20]
			*Eurotium*	[22]
		antioxidative activity	*E. rubrum*	[23]
2	physcion	cytotoxic activity antiviral activity	*E. herbariorum NU-2*	[16]
			*E. chevalieri KUFA 0006*	[20]
			*E. repens*	[24]
			*E. chevalieri MUT 2316*	[25]
3	erythroglaucin	antimicrobial activity	*Eurotium*	[22]
			*E. cristatum* KUFC 7356	[26]
4	emodin	antimicrobial activity cytotoxic activity	*E. chevalieri* KUFA 0006	[20]
			*Eurotium*	[22]
			*E. rubrum*	[27]
			*E. cristatum* KUFC 7356	[26]
			*E. rubrum*	[28]
5	catenarin		*E. herbariorum* NU-2	[16]
			*Eurotium*	[22]
			*E. rubrum*	[27]
			*E. cristatum* KUFC 735	[26]
6	rubrocristin		*Eurotium*	[22]
			*E. rubrum*	[27]
7	rubrocristin-8-methylether		*Eurotium*	[22]
8	rubrocristin-6-acetate		*Eurotium*	[22]
9	querstin-6-methylether		*Eurotium*	[22]
10	2-*O*-methyleurotinone	antioxidative activity	*E. echinulatum*	[29]
			*E. rubrum*	[23]
11	2,12-dimethyleurotinone		*E. echinulatum*	[29]
12	eurotinone		*E. echinulatum*	[29]
13	physcion-10,10′-bianthrone		*E. herbariorum* NU-2	[16]
14	questinol	anti-inflammatory activity	*Eurotium* sp. M30 XS-2012	[11]
			*E. herbariorum* NU-2	[16]
			*E. chevalieri* KUFA 0006	[20]
			*E. amstelodami*	[30]
15	asperflavin	antioxidative activity cytotoxic activity antimicrobial activity anti-inflammatory activity	*Eurotium* sp. M30 XS-2012	[11]
			*E. herbariorum* NU-2	[16]
			*E. rubrum*	[23]
			*E. cristatum* EN-220	[31]
			*E. repens*	[24]
			*E. chevalieri* MUT 2316	[25]
			*E. amstelodami*	[32]
16	variecolorquinone A		*Eurotium* sp. M30 XS-2012	[26]
			*E. cristatum* EN-220	[31]
17	2-*O*-methyl-4-*O*-(*α*-_D_-ribofuranosyl)-9-dehydroxyeurotinone	antioxidative activity	*E. rubrum*	[23]
18	2-*O*-methyl-9-dehydroxyeurotinone	antioxidative activity	*E. rubrum*	[23]
19	eurorubrin	antioxidative activity antimicrobial activity insecticidal activity	*E. rubrum*	[23]
			*E. cristatum* EN-220	[31]
20	3-*O*-(*α*-_D_-ribofuranosyl)-questin	antioxidative activity	*E. rubrum*	[23]
			*E. cristatum* EN-220	[31]
21	3-*O*-(*α*-_D_-ribofuranosyl)-questinol	antimicrobial activity	*E. cristatum* EN-220	[31]
22	asperflavin ribofuranoside		*E. cristatum* EN-220	[31]
23	9-dehydroxyeurotinone	cytotoxic activity antimicrobial activity	*E. rubrum*	[28]
24	acetylquestinol		*E.chevalieri* KUFA 0006	[20]
25	rubrumol	cytotoxic activity	*E. rubrum*	[27]
Benzaldehyde derivatives				
26	2-(2′,3-epoxy-1′-heptenyl)-6-hydroxy-5-(3″-methyl-2″-butenyl)-benzaldehyde		*E. rubrum*	[33]
27	(*E*)-6-hydroxy-7-(3-methyl-2-butenyl)-2-(3-oxobut-1-enyl)-chroman-5-carbaldehyd		*E. rubrum*	[33]
28	2-(1′,5′-heptadienyl)-3,6-dihydroxy-5-(3″-methyl-2″-butenyl)-benzaldehyde		*E. rubrum*	[33]
29	eurotirumin	cytotoxic activity	*E. rubrum*	[33]
			*Eurotium* sp. SCSIO F452	[34]
30	chaetopyranin		*E. rubrum*	[33]
31	flavoglaucin	antioxidative activity antimicrobial activity antimalarial activity anti-inflammatory activity cytotoxic activity	*Eurotium*	[35]
			*E. cristatum*	[14]
			*E. repens*	[19]

			*Eurotium* sp. SF-5989	[36]
			*E. rubrum*	[33]
			*E. repens*	[37]
			*E. cristatum*	[38]
32	aspergin		*E. rubrum*	[33]
33	isotetrahydroauroglaucin	anti-inflammatory activity	*Eurotium* sp. SF-5989	[36]
			*E. rubrum*	[33]
34	isodihydroauroglaucin	antioxidative activity	*Eurotium*	[35]
			*E. cristatum*	[14]
			*E. rubrum*	[33]
			*E. repens*	[37]
35	2-(2′,3-epoxy-1′,3′-heptadienyl)-6-hydroxy-5-(3-methyl-2-butenyl)-benzaldehyde	antimicrobial activity antimalarial activity	*E. cristatum*	[14]
			*E. repens*	[19]

			*E. rubrum*	[33]
			*E. repens*	[37]
36	2-(2′,3-epoxy-1′,3′,5′-heptatrienyl)-6-hydroxy-5-(3-methyl-2-butenyl)-benzaldehyde		*E. rubrum*	[33]
			*E. cristatum*	[14]
37	eurotirubrin A		*E. rubrum*	[39]
38	eurotirubrin B		*E. rubrum*	[39]
39	auroglaucin	antioxidative activity antimicrobial activity antimalarial activity	*Eurotium*	[35]
			*E. repens*	[19]

40	tetrahydroauroglaucin	antioxidative activity antimicrobial activity antimalarial activity cytotoxic activity	*Eurotium*	[35]
			*E. cristatum*	[14]
			*E. repens*	[19]

			*E. repens*	[37]
			*E. repens*	[24]
41	dihydroauroglaucin	antioxidative activity antimicrobial activity antiviral activity	*Eurotium*	[35]
			*E. cristatum*	[14]
			*E. repens*	[19]
			*E. repens*	[37]
			*E. chevalieri* MUT 2316	[25]
42	(*E*)-2-(hept-1-enyl)-3-(hydroxymethyl)-5-(3-methylbut-2-enyl)-benzene-1,4-diol	antimicrobial activity antimalarial activity	*E. repens*	[19]
			*E. repens*	[37]
43	(*E*)-4-(hept-1-enyl)-7-(3-methylbut-2-enyl)-2,3-dihydrobenzofuran-2,5-diol		*E. repens*	[37]
44	(3′S*,4′R*)-6-(3′,5-epoxy-4′-hydroxy-1′-heptenyl)-2-hydroxy-3-(3″-methyl-2″-butenyl)-benzaldehyde		*Eurotium*	[40]
45	3′-OH-tetrahydroauroglaucin		*Eurotium*	[40]
46	cristaldehyde A	anti-inflammatory activity	*E. cristatum*	[38]
47	cristaldehyde B		*E. cristatum*	[38]
Indole diketopiperazine alkaloids				
48	echinulin	antimicrobial activity antioxidative activity insecticidal activity	*E. cristatum*	[26]
			*E. cristatum* EN-220	[41]
			*E. cristatum*	[42]
			*E. repens*	[43]
			*E. repens*	[24]
			*E. amstelodami*	[44]
			*E. rubrum*	
			*E. herbariorum*	
			*E. cristatum* EN-220	[45]
			*Eurotium* sp. SCSIO F452	[46]
			*Eurotium*	[47]
49	neoechinulin A	antimicrobial activity antioxidative activity insecticidal activity anti-inflammatory activity	*Eurotium* sp. M30 XS-2012	[11]
			*E. cristatum*	[26]
			*E. cristatum* EN-220	[41]
			*E. cristatum*	[42]
			*E. rubrum* Hiji 025	[48]
			*E. amstelodami*	[44]
			*E. rubrum*	
			*E. herbariorum*	
			*E. rubrum* MA-150	[49]
			*Eurotium*	[47]
			*Eurotium* sp. *SF-5989*	[50]
50	neoechinulin B	antioxidative activity	*E. herbariorum* NU-2	[16]
			*E. amstelodami*	[44]
			*E. rubrum.*	
			*E. herbariorum*	
			*E. cristatum* EN-220	[45]
			*Eurotium* sp. *SCSIO F452*	[46]
51	preechinulin		*E. cristatum* EN-220	[41]
			*E. amstelodami*	[44]
			*E. rubrum*	
			*E. herbariorum*	
52	neoechinulin E	insecticidal activity antioxidative activity	*E. cristatum*	[26]
			*E. amstelodami*	[44]
			*E. herbariorum*	
			*E. rubrum* MA-150	[49]
			*E. rubrum*	[51]
53	7-*O*-methylvariecolortide A	caspase-3 inhibitory activity	*E. rubrum*	[52]
			*Eurotium*	[53]
54	variecolortide A		*E. rubrum*	[52]
55	variecolortide B	caspase-3 inhibitory activity	*E. rubrum*	[52]
			*E. rubrum* MA-150	[49]
			*Eurotium*	[53]
56	variecolortide C	caspase-3 inhibitory activity	*E. rubrum*	[52]
			*E. rubrum* MA-150	[49]
			*Eurotium*	[53]
57	fructigenine A		*Eurotium* sp. SF-5130	[54]
58	eurocristatine		*E. cristatum*	[26]
59	variecolorin J		*E. cristatum*	[26]
			*E. rubrum*	[28]
60	12-demethyl-12-oxo-eurotechinulin B	cytotoxic activity	*E. rubrum*	[28]
61	variecolorin G	cytotoxic activity insecticidal activity antioxidative activity	*E. rubrum*	[28]
			*E. cristatum* EN-220	[41]
			*E. rubrum* MA-150	[49]
			*Eurotium* sp. SCSIO F452	[46]
62	eurotechinulin B		*E. rubrum*	[28]
63	cryptoechinuline G		*E. rubrum*	[28]
64	alkaloid E-7	cytotoxic activity insecticidal activity	*E. rubrum*	[28]
			*E. cristatum* EN-220	[45]
65	isoechinulin B	antioxidative activity	*E. herbariorum* NU-2	[16]
			*E. rubrum 31*	[28]
66	cristatumin A	antimicrobial activity	*E. cristatum* EN-220	[41]
67	cristatumin B	insecticidal activity	*E. cristatum* EN-220	[41]
68	cristatumin C		*E. cristatum* EN-220	[41]
69	cristatumin D	antimicrobial activity	*E. cristatum* EN-220	[41]
70	isoechinulin A	antioxidative activity insecticidal activity	*E. herbariorum* NU-2	[16]
			*E. cristatum* EN-220	[41]
			*E. rubrum* MA-150	[49]
			*Eurotium* sp. SCSIO F452	[46]
71	tardioxopiperazine A	antimicrobial activity	*E. cristatum* EN-220	[41]
72	cristatumin E	antimicrobial activity cytotoxic activity	*E. herbariorum* HT-2	[55]
73	rubrumazine A		*E. rubrum* MA-150	[49]
74	rubrumazine B	insecticidal activity	*E. rubrum* MA-150	[49]
			*E. cristatum* EN-220	[45]
75	rubrumazine C		*E. rubrum* MA-150	[49]
76	dehydroechinulin	insecticidal activity antioxidative activity	*E. cristatum*	[42]
			*E. rubrum* MA-150	[49]
			*E. cristatum* EN-220	[45]
			*Eurotium* sp. SCSIO F452	[46]
77	variecolorin E		*E. rubrum* MA-150	[49]
78	dihydroxyisoechinulin A	antimicrobial activity	*Eurotium* sp. M30 XS-2012	[11]
			*E. rubrum* MA-150	[49]
79	variecolorin L		*E. rubrum* MA-150	[49]
80	tardioxopiperazine B		*E. rubrum* MA-150	[49]
81	_L_-alanyl-_L_-tryptophan anhydride	antimicrobial activity	*Eurotium* sp. M30 XS-2012	[11]
			*E. rubrum* MA-150	[49]
82	cristatumin F		*E. cristatum*	[42]
83	variecolorin O	antioxidative activity	*E. herbariorum* NU-2	[16]
			*E. cristatum*	[42]
			*Eurotium* sp. SCSIO F452	[46]
84	*N*-(4′-hydroxyprenyl)-cyclo(alanyltryptophyl)		*E. cristatum* EN-220	[45]
85	isovariecolorin I	insecticidal activity	*E. cristatum* EN-220	[45]
86	30-hydroxyechinulin		*E. cristatum* EN-220	[45]
87	29-hydroxyechinulin		*E. cristatum* EN-220	[45]
88	rubrumline M		*E. cristatum* EN-220	[45]
89	neoechinulin C	insecticidal activity	*E. cristatum* EN-220	[45]
90	didehydroechinulin	insecticidal activity	*E. cristatum* EN-220	[45]
91	variecolorin H		*E. cristatum* EN-220	[45]
92	(11*R*,14*S*)-3-(1*H*-indol-3ylmethyl)6-isopropyl-2,5-piperazinedione		*E. chevalieri* KUFA 0006	[20]
93	variecolortin A	antioxidative activity	*Eurotium* sp. SCSIO F452	[56]
94	variecolortin B	cytotoxic activity	*Eurotium* sp. SCSIO F452	[56]
95	variecolortin C	cytotoxic activity	*Eurotium* sp. SCSIO F452	[56]
96	eurotiumin A	antioxidative activity	*Eurotium* sp. SCSIO F452	[46]
97	eurotiumin B	antioxidative activity	*Eurotium* sp. SCSIO F452	[46]
98	eurotiumin C	antioxidative activity	*Eurotium* sp. SCSIO F452	[46]
99	fintiamin		*Eurotium*	[57]
Other compounds				
100	chevalone A		*E. chevalieri*	[58]
101	chevalone B	cytotoxic activity	*E. chevalieri*	[58]
102	chevalone C	antimicrobial activity cytotoxic activity	*E. chevalieri*	[58]
103	chevalone D	antimalarial activity cytotoxic activity	*E. chevalieri*	[58]
104	aszonapyrone A		*E. chevalieri*	[58]
105	aszonapyrone B		*E. chevalieri*	[58]
106	CJ-12662	antimalarial activity antimicrobial activity cytotoxic activity	*E. chevalieri*	[58]
107	3*β*,5*α*-dihydroxy-10*α*-methyl-6*β*-acetoxy-ergosta-7,22-diene		*E.rubrum*	[59]
108	3*β*,5*α*-dihydroxy-6*β*-acetoxyergosta-7,22-diene		*E.rubrum*	[59]
109	(22*E*,24*R*)-ergosta-7,22-dien-3*β*-ol		*E.rubrum*	[59]
110	(22*E*,24*R*)-ergosta-7,22-dien-6*β*-methoxy-3*β*,5*α*-diol		*E.rubrum*	[59]
111	(22*E*,24*R*)-ergosta-7,22-dien-3*β*,5*α*,6*β*-triol		*E.rubrum*	[59]
112	(22*E*,24*R*)-ergosta-7,22-dien-3*β*,5*α*,6*α*-triol		*E.rubrum*	[59]
113	(22*E*,24*R*)-3*β*,5*α*,9*α*-trihydroxyergosta-7,22-dien-6-one		*E.rubrum*	[59]
114	(22*E*,24*R*)-3*β*,5*α*-dihydroxyergosta-7,22-dien-6-one		*E.rubrum*	[59]
115	(22*E*,24*R*)-5*α*,8*α*-epidioxyergosta-6,22-dien-3*β*-ol		*E.rubrum*	[59]
116	(22*E*,24*R*)-5*α*,8*α*-epidioxyergosta-6,22-dien-3*β*-acetate		*E.rubrum*	[59]
117	(22*E*,24*R*)-ergosta-4,6,8(14),22-tetraen-3-one		*E.rubrum*	[59]
118	euroticin A		*Eurotium* sp. SCSIO F452	[15]
119	euroticin B	antioxidative activity	*Eurotium* sp. SCSIO F452	[15]
120	euroticin C	antioxidative activity cytotoxic activity	*Eurotium* sp. SCSIO F452	[60]
121	euroticin D		*Eurotium* sp. SCSIO F452	[60]
122	euroticin E		*Eurotium* sp. SCSIO F452	[60]
123	euroticin F	cytotoxic activity antioxidative activity	*Eurotium* sp. SCSIO F452	[34]
124	euroticin G	antioxidative activity *α*-glucosidase inhibitory activity	*Eurotium* sp. SCSIO F452	[34]
125	euroticin H	cytotoxic activity *α*-glucosidase inhibitory activity	*Eurotium* sp. SCSIO F452	[34]
126	euroticin I	cytotoxic activity	*Eurotium* sp. SCSIO F452	[34]
127	salicylaldehydium A	cytotoxic activity	*Eurotium* sp. SCSIO F452	[61]
128	salicylaldehydium B		*Eurotium* sp. SCSIO F452	[61]
129	asperglaucin A	antimicrobial activity	*Aspergillus chevalieri* SQ-8	[17]
130	asperglaucin B	antimicrobial activity	*Aspergillus chevalieri* SQ-8	[17]
131	citrinin		*Eurotium*	[62]
132	ochratoxin A		*Eurotium*	[62]
133	gliotoxin		*Eurotium*	[62]
134	aflatoxins		*Eurotium*	[62]
135	sterigmatocystin		*Eurotium*	[62]
136	cyclopenol		Eurotium sp. SF-5130	[54]
137	mycophenolic acid		*E. repens*	[63]
138	2-(2-methyl-3-en-2-yl)-1*H*-indole-3-carbaldehyde		*E. chevalieri* KUFA 0006	[20]
139	(2,2-dimethylcyclopropyl)-1*H*-indole-3-carbaldehyde		*E. chevalieri* KUFA 0006	[20]
140	2-(1,1-dimethyl-2-propen-1-yl)-1*H*-indole-3-carboxaldehyde		*Eurotium* sp. SCSIO F452	[64]
141	ergosterol		*E. chevalieri*	[58]
142	2[(2,2-dimethylbut-3-enoyl)amino]benzoic acid		*E. chevalieri* KUFA 0006	[20]
143	6,8-dihydroxy-3-(2-hydroxypropyl)-7-methyl-1*H*-isochromen-1-one		*E. chevalieri* KUFA 0006	[20]
144	ergosterol 5,8-endoperoxide		*E. chevalieri* KUFA 0006	[20]
145	(11*S*,14*R*)-cyclo(tryptophylvalyl)		*E. chevalieri* KUFA 0006	[20]
146	cinnalutein		*E. chevalieri* MUT 2316	[25]
147	*cyclo*-_L_-Trp-_L_-Ala		*E. chevalieri* MUT 2316	[25]
148	eurochevalierine	antimalarial activity antimicrobial activity cytotoxic activity	*E.* chevalieri	[58]
149	sequiterpene		*E.* chevalieri	[58]
150	zinniol		*E.rubrum* SH-823	[65]
151	butyrolactone I		*E.rubrum* SH-823	[65]
152	aspernolide D		*E.rubrum* SH-823	[65]
153	vermistatin		*E.rubrum* SH-823	[65]
154	methoxyvermistatin		*E.rubrum* SH-823	[65]
155	eurothiocin A	*α*-glucosidase inhibitory activity	*E.rubrum* SH-823	[65]
156	eurothiocin B	*α*-glucosidase inhibitory activity	*E.rubrum* SH-823	[65]
157	7-isopentenylcryptoechinuline D		*E.rubrum*	[28]
158	methyl linoleate		*Eurotium* sp. SCSIO F452	[64]
159	*cyclo*-(_L_-Pro-_L_-Phe)		*Eurotium* sp. SCSIO F452	[46]
160	eurotinoid A	antioxidative activity	*Eurotium* sp. SCSIO F452	[66]
161	eurotinoid B	antioxidative activity	*Eurotium* sp. SCSIO F452	[66]
162	eurotinoid C	antioxidative activity	*Eurotium* sp. SCSIO F452	[66]
163	dihydrocryptoechinulin D	cytotoxic activity antioxidative activity	*Eurotium* sp. SCSIO F452	[66]
164	eurotone A		*Eurotium* sp. SCSIO F452	[67]
165	5,7-dihydroxy-4-methylphthalide	antimicrobial activity	*E. repens* *E. repens*	[19] [37]
166	cristatumside A		*E. cristatum* EN-220	[31]
167	eurotiumide A	insecticidal activity	*Eurotium* sp. XS-200900E6	[21]
168	eurotiumide B	insecticidal activity	*Eurotium* sp. XS-200900E6	[21]
169	eurotiumide C	insecticidal activity	*Eurotium* sp. XS-200900E6	[21]
170	eurotiumide D	insecticidal activity	*Eurotium* sp. XS-200900E6	[21]
171	eurotiumide E		*Eurotium* sp. XS-200900E6	[21]
172	eurotiumide F		*Eurotium* sp. XS-200900E6	[21]
173	eurotiumide G		*Eurotium* sp. XS-200900E6	[21]
174	viridicatol		*Eurotium* sp. SF-5130	[54]
175	monacolin K		*E. cristatum*	[68]
176	cristaquinone A	anti-inflammatory activity	*E. cristatum*	[38]
177	6-*O*-*α*-_D_–ribofuranoside		*E. cristatum* EN-220	[31]

## Data Availability

No new data were created or analysed in this study. Data sharing is not applicable to this article.

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
