# Peer review of "Secondary Metabolites from the Genus Eurotium and Their Biological Activities"

_foods, 2023, doi:10.3390/foods12244452_

Round 1

Reviewer 1 Report

Comments and Suggestions for Authors

Authors tried to review the secondary metabolites in Eurotium.

Major comments

1.    Please provide higher resolution of all figures.

2.    This manuscript is merely accumulation of previous data without reviewing. Please add more review or perspective. If possible, please provide metabolism of mentioned anthraquinones, benzaldehyde derivatives, indole diketopiperazine alkaloids and etc.

Comments on the Quality of English Language

Quality of English looks fine.

Author Response

Response to Reviewer

Thank you very much for taking the time to review this manuscript. Please find the detailed responses below and the corresponding revisions in the re-submitted files.

Comments 1: Please provide higher resolution of all figures.

Response 1: Thank you for pointing this out. We agree with this comment. Therefore, we've updated the images with more clarity, so please check them out!

Comments 2: This manuscript is merely accumulation of previous data without reviewing. Please add more review or perspective. If possible, please provide metabolism of mentioned anthraquinones, benzaldehyde derivatives, indole diketopiperazine alkaloids and etc.

Response 2: Thank you for pointing this out. this review provided the recent advances in the secondary metabolites and their bioactivities in the genus Eurotium for the first time. Unfortunately, however, there are very few studies related to the metabolic profile you described, so we add in the conclusion section the prospect that more research is needed on the pathways of formation of secondary metabolites of Eurotium.

Reviewer 2 Report

Comments and Suggestions for Authors

Dear Editor and Authors,

In this review article entitled "Secondary Metabolites from Genus Eurotium and Their Biological Activities" the authors describe the recent advances in the secondary metabolites and their bioactivities in the genus Eurotium. At first sight, the article is not badly prepared, it is readable and clear, the tables and figures (except two) are clear and understandable, also the text of this review looks good, but in my opinion it is too short - it contains too few literary sources! In particular, I recommend revising the introduction to the article and adding some relevant information. Below are comments on the article which, once corrected, will make it possible to edit the article in the future: 

1.      In the article, I also miss the clearly stated customs and novelties!

2.      Introduction: It would certainly be appropriate to mention in the article about the genus Eurotium that it has been renamed to the genus Aspergillus according to the new nomenclature and that the genus Eurotium is a teleomorpf of the genus Aspergillus, but this is very confusing and therefore most mycologists stick to common and already used names (10.3852/14-060), but important mycologists like Pitt should be quoted Hocking, Samson etc... In addition, the article also cites the author Hubka et al. (2013), who also address this issue in this particular article! Add other relevant information, especially from recent years!

3.      As the genera Aspergillus and Eurotium are mainly food spoiling and degrading fungi, it would also be appropriate to make a connection, since the authors want to publish an article, which in my opinion is too mycological and more suitable for the journal JoF, in the journal FOODS, whose website states this: Foods (ISSN 2304-8158) is an international, peer-reviewed, open access scientific journal that provides an advanced forum for studies related to all aspects of food research, with an emphasis on the "science of food".

4.      Check throughout the text for Latin names that must be written in italics, e.g. ....L13 Aspergillus

5.      L281.. why are terpenoids marked with a link?

6.      The quality of images 3 and 4 is poor...blurred numbers and labels, please replace the images with better quality and higher resolution.

7.      Throughout the article you should correct the literature and quote it correctly!!! For example, L261, 266 or 274 (but there are these errors before and after in the text) when quoting an author, write the author first, followed immediately by the number assigned to him, for example Zhong et al. (13) etc... Follow the instructions of the journal FOODS!

8.      It is also necessary to correct the cited literature, which also does not meet the requirements of the journal! For example, years must be in bold, etc...

9.      Conclusion: I don't understand why the Table 1 is placed after the Conclusion section and mentioned first in conclusion section??? I recomended move Table 1 somewhere else, for example in the extended introduction, because it looks chaotic in the conclusion! And the conclusions of the work need to be better defined

10.   The goal of the work and novelties must be clarified

11.   The article must be in accordance with the requirements (aim and scope) of FOODS journal

Comments on the Quality of English Language

I would recommend extensive editing of English language. There are a lot of mistakes in the text, especially in the prepositions and in some of the sentence structures.

Author Response

Dear reviewer, we have revised the manuscript according to your request, and the specific answers are in the attachment.

Round 2

Reviewer 1 Report

Comments and Suggestions for Authors

No comments

Author Response

Response to reviewers

Thank you very much for taking the time to review this manuscript. Please find the following detailed responses and corresponding changes in the resubmission.
You do not seem to have given specific comments on my revised manuscript, and in response to your scoring, we have revised our English writing even further in the hope of meeting your requirements!

Reviewer 2 Report

Comments and Suggestions for Authors

Dear Editor and authors,

The authors have improved their manuscript considerably and I thank them for their detailed answers. Nevertheless, I have a few minor comments on the article.

-All names in vivo, in vitro, in situ, in silico must be in italics! Correct throughout.
- The authors of the article added and corrected the mentioned errors, which I appreciate, but there is still a lot of information missing with the connection between the topic of the article and the content focus of the journal FOODS (a tea is very little), try to focus especially on spoilage foods with characterized by this genus (for example, foods with a lower aw ) and add more information where the metabolites of the genus Eurotium are used in the food industry.
-The authors stated that they moved Table 1 to the beginning, but I still see it at the end - please fix it.

Comments on the Quality of English Language

Moderate editing of English language is required. I keep spotting grammatical errors, spelling mistakes, extra characters or the wrong words in the article.

Author Response

Dear reviewers
I have followed your request and added the link between our article and food in the introduction, text and conclusion sections.
In addition, we have corrected grammatical errors throughout the article and moved Table 1 to the back of the introduction as you requested.
Thank you very much for your correction and good luck!